# Cu-Doped ZnO Electronic Structure and Optical Properties Studied by First-Principles Calculations and Experiments

**DOI:** 10.3390/ma12010196

**Published:** 2019-01-08

**Authors:** Zhanhong Ma, Fengzhang Ren, Xiaoli Ming, Yongqiang Long, Alex A. Volinsky

**Affiliations:** 1School of Materials Science and Engineering, Henan University of Science and Technology, Luoyang 471023, China; mazhanhong@163.com (Z.M.); mingxiaoli0218@126.com (X.M.); yqlong@mail.haust.edu.cn (Y.L.); 2Henan Collaborative Innovation Centre of Non-Ferrous Generic Technology, Luoyang 471023, China; 3Department of Mechanical Engineering, University of South Florida, Tampa, FL 33620, USA; volinsky@usf.edu

**Keywords:** Cu doped ZnO, first-principles calculations, electronic structure, absorption spectrum

## Abstract

The band structure, the density of states and optical absorption properties of Cu-doped ZnO were studied by the first-principles generalized gradient approximation plane-wave pseudopotential method based on density functional theory. For the Zn_1-*x*_Cu*_x_*O (*x* = 0, *x* = 0.0278, *x* = 0.0417) original structure, geometric optimization and energy calculations were performed and compared with experimental results. With increasing Cu concentration, the band gap of the Zn_1_*-_x_*Cu*_x_*O decreased due to the shift of the conduction band. Since the impurity level was introduced after Cu doping, the conduction band was moved downwards. Additionally, it was shown that the insertion of a Cu atom leads to a red shift of the optical absorption edge, which was consistent with the experimental results.

## 1. Introduction

ZnO is a direct band gap n-type semiconductor with an exciton binding energy of 60 meV and a band gap of 3.37 eV at room temperature [1,2,3]. Due to the fine crystal grains, nano-ZnO changes its crystal structure and surface electronic structure, resulting in surface, volume, macroscopic tunneling and quantum size effects, as well as high dispersion and high transparency, which are not found in macroscopic ZnO particles. Therefore, many scholars have studied the synthesis, characterization and applications of nano-ZnO [4,5]. In recent years, ordinary ZnO has found many uses and special functions in optics, magnetism, and mechanics, and there is potential for a wide use as photocatalyst in solar energy applications, battery, UV laser, and lithium ion battery materials. Since half of the tetrahedral voids of the wurtzite structure of ZnO are not filled, the structure is relatively open, so ZnO can have inherent defects. As an electronic material, ZnO is extremely important for the control of its inherent defects which control its properties. Properties of ZnO doped with transition metal atoms, such as copper, are tunable to suit specific needs and applications. Doping can cause dramatic changes in electronic, optical, and magnetic properties by changing the electronic structure of ZnO [6,7,8,9]. For copper doping, studies of Chakraborty et al. [10] have shown that 5%–10% Cu-doped ZnO has a lower band gap value than undoped ZnO. Lower doping concentration resulted in a blue absorption edge shift compared to the undoped ZnO. Copper doping reduces the recombination of electrons and holes, and the band gap value decreases from 3.21 eV (zinc oxide) to 3.07 eV. Kayani [11] showed that the crystallite size of the film increased as the doping ratio of Cu increased from 2 wt % to 10 wt %. Increase in Cu doping leads to an insignificant decrease in the optical band gap of the thin films. Slimi [12] found that the crystallite sizes of ZnO and Cu-doped ZnO nanoparticles were within the 32–38 nm range and the Cu dopant uniformly substituted Zn positions. Optical absorption spectra showed a red shift of the absorption edge due to the merging of the impurity band into the conduction band (CB). As the Cu content increases, the visible photoluminescence intensity decreases due to the scattering of the excitation radiation by surface-adsorbed dopant atoms and the non-radiative recombination process introduced by Cu impurities. The results of Fang [13] show that Cu doping did not lead to the formation of a secondary phase, but slightly reduced the particle size. The valence state of Cu in ZnO was confirmed to be +2. With the increase of Cu doping concentration, the photoluminescence intensity decreased under the 325 nm excitation wavelength. Das [14] calculated that the band gap (E_g_) of ZnO decreases with Cu doping, which can be attributed to the sp-d exchange interaction between the ZnO band electrons and localized d electrons of Cu^2+^ ions. Zheng et al. [15] calculated the density of Cu-doped ZnO by the modified Becke and Johnson potential (MBJ)-Coherent-Potential Approximation (CPA) method. It was found that the impurity level of 0.1% copper element is above the Fermi level, indicating that Cu-doped ZnO is a p-type semiconductor.

In summary, for Cu doping, the theoretical and experimental research results are separated in the literature, and the linking of experimental results with theory is lacking. At the same time, for copper doping, the first-principles calculations focus on magnetic properties and the calculation of optical properties is rare. This paper is aimed at this problem, the combining of experimental results with theoretical analysis. The effects of Cu doping on the electronic structure and optical properties of ZnO were investigated. By comparing experimental and theoretical differences, the theoretical basis for improving the photoelectric performance of doped ZnO is proposed.

## 2. Theoretical Methods and Experimental Details

### 2.1. Thoretical Methods

The crystal structure of wurtzite ZnO belongs to the hexagonal system, the space group is P63mc, and the symmetry is C46v. In the crystal structure, each zinc ion is surrounded by four oxygen ions to form [ZnO_4_], and the unit cell parameters are *a* = 0.3249 nm, *c* = 0.5205 nm, *α* = *β* = 90° *γ* = 120°.

In the experiment, the molar ratio of copper doping was chosen to be between 0 and 5. In order to correspond with the experimental results, the model constructed in this paper is a pure ZnO (1 × 1 × 1) unit cell model; Cu_0.0417_Zn_0.9583_O (2 × 2 × 3) with a Cu atom replacing Zn atom (corresponding doping amount is 4.2%), Cu_0.0278_Zn_0.9722_O (3 × 3 × 2) with a Cu atom replacing Zn atom (corresponding doping amount 2.8%); and the crystal structure is shown in Figure 1.

The calculations were performed using density functional theory (DFT), as implemented in the Castep software in the Materials Studio package [16]. The exchange-related energy can describe the selected potential as a super soft potential with the Perdew–Burke–Ernzerhof (PBE) functional (ultra-soft Pseudopotentials, USP) [16,17]. The valence electronic configurations used for the construction were Zn 3d^10^4s^2^, Cu 3d^10^4s^1^, and O 2s^2^2p^4^, respectively, and the calculations were performed in the reciprocal space. At the end of the self-consistent process, the overall energy of the structure converges to 2 × 10^−5^ eV/atom, the force on each atom is less than 0.05 eV/nm, the tolerance offset is 0.0002 nm, and the stress deviation is 0.1 GPa. The K points in the Brillouin zone are set to 9 × 9 × 6 (1 × 1 × 1 unit cells); 3 × 3 × 2 (3 × 3 × 2 supercell); 4 × 2 × 2 (2 × 2 × 3) supercell). In the calculations, the crystal structure is optimized first, and the electronic structure and optical properties are calculated according to the obtained structural parameters. The traditional density functional theory (DFT) calculations (Local density approximation (LDA) or The generalized gradient approximation (GGA) cannot be accurately used to describe the transition group elements of the d electrons and the oxides of the localized oxygen element p electrons. In some cases, we can replace the LDA or GGA calculation by introducing a strong correlation term between atoms, which is described in the model by the Hubbard parameter U (repulsive energy), called the LDA + U or GGA + U method. The GGA + U accurately describes the electronic structure of oxides doped with transition metals and localized oxygen elements [18,19]. When calculating energy, the electrons are treated by spin polarization. Since the traditional GGA method calculates the electronic structure and underestimates the band gap width of the metal oxide, the calculated band gap of pure ZnO is 0.740 eV, which is close to other calculations [20,21,22]. However, it is much less than the experimental value (3.37 eV), which is due to the limitation of DFT in GGA. This paper uses the GGA + U method to correct the band gap [23,24,25,26]. After several iterations, it was found that in all the systems the 3d state of Zn and the 2p state of O take U = 5 eV and U = 8 eV, respectively. The band gap of zinc oxide is 3.373 eV, which is very close to the theoretical value. Based on the optimization of the geometric structure, the electronic structure and absorption spectrum distribution of all models were calculated. Finally, the results were analyzed and discussed.

### 2.2. Experimental Details

ZnO nano powders were synthesized as follows. First, commercial zinc powders (99% pure) were cleaned in acetone, ethanol, and distilled water sequentially by an ultrasonic machine (SYS5200, Kunshan ultrasonic instrument co., LTD, Kunshan, China). Then, 4 g of cleaned zinc powders was immersed in 1.0 mol/l oxalic acid aqueous solution (50 mL) for various times during the precursor synthesis step. Next, the reaction products in the solution were poured onto filter papers and dried at ambient conditions to obtain white precursor powders. The 1.0 g of the precursor was separated into different volumes of 0.005 mol/L copper nitrate solution, dispersed by ultrasonic vibration for 45 min, and left at room temperature for 24 h. Then the precipitate was dried and poured into an alumina crucible, and calcined at 500 °C for 2 h in a high-temperature energy-saving box furnace. The dried powders in the alumina crucible were heated in a SYS-G-Z-13 tube furnace in air at 500 °C for 2 h. The crystal structures of as-prepared samples were examined with a D8-Advanced X-ray diffractometer (Bruker Corporation, Karlsruhe, Germany), using 40 kV, 30 mA, Cu Kα X-rays. The light absorption performance of the as-prepared samples was characterized with an ultraviolet-visible absorption spectrometer (UV-2600, Shimadzu, Tokyo, Japan).

## 3. Results and Discussion

### 3.1. Lattice Structure and Stability Analysis

In order to study the influence of copper doping on the structure and stability, geometrical structure optimization calculations were carried out for the three models. The results show the lattice parameters of each system listed in Table 1.

The Cu-doped system stability was estimated using the defect formation energy *E*_f_. *E*_f_ is defined as follows:
*E*_f_ = *E*_ZnO,Cu_*− E*_ZnO_ + *E*_Zn_ − *E*_Cu_(1)
where *E*_ZnO,Cu_ is the DFT total energy of the doped supercell, *E*_ZnO_ is the energy of the supercell without impurities, and *E_Zn_* and *E_Cu_* are the total energies of the bulk Zn and Cu metals per atom, respectively.

It can be seen from Table 1 that the optimized lattice parameters of pure zinc oxide are *a* = *b* = 0.3249 nm and *c* = 0.5205 nm, respectively, which is consistent with the experimental results [27]. In this paper, copper was doped in the zinc oxide, and there are two valence states of copper. In the literature [12], the copper in zinc oxide is divalent, and the radius of Cu^2+^ (0.073 nm), and the radius of the Zn^2+^ (0.074 nm) are very close, so that the change of the lattice constant is small, and the ZnO material does not undergo significant lattice distortion. This reduces the internal stress of the film, indirectly improving the quality of the film, which is consistent with experimental reports [14].

It can be seen from Table 1 that the higher doping concentration, the higher is the formation energy of the system after doping. This indicates that the initiation of doping of copper leads to an increase in the free energy of the system and a decrease in stability. As the amount of doping increases, the system gradually stabilizes.

In this paper, Cu-doped ZnO was prepared by a two-step method. The doping molar ratio of Cu was 3%, the calcination temperature was 500 °C, and the calcination time was 3 h. Figure 2 shows the XRD pattern of copper-doped zinc oxide. Compared with the ZnO standard map of pure wurtzite, the structure of zinc oxide after Cu doping is a wurtzite structure, and the doping does not change the symmetry of the crystal structure. According to the Scherrer formula, the particle size of Cu-doped ZnO is 22 nm. There is no copper peak formed after doping because the doped copper enters the zinc oxide structure. The unit cell parameters of ZnO after doping are *a* = *b* = 0.32498 nm, *c* = 0.52066 nm, which is consistent with the calculation results.

### 3.2. Doping Effects on the ZnO Band Structure and Density of Electronic States

Using the GGA + U method, U^d^ of Zn, U^p^ anO and U^d^ of Cu in the pure ZnO system were taken as 5 eV, 8 eV, 2.5 eV respectively. The energy band distribution and the density of states distribution of the pure ZnO system are shown in Figure 3 and Figure 4, respectively. It can be seen from Figure 3 that the top of the valence band and the bottom of the conduction band are at the high symmetry point (G point), and the top of the valence band (VB) is at the energy zero point (i.e., the Fermi level), indicating pure ZnO. The semiconductor is a direct band gap semiconductor. The band gap is the distance from the highest point of the valence band to the lowest point of the conduction band. In Figure 3, the band gap of zinc oxide is 3.373 eV, which is very close to the theoretical value. This indicates that it is reasonable to use 5 eV and 8 eV for U^d^, Zn and U^p^, O in the pure ZnO supercell, respectively. The default Fermi level of the software is zero energy (the same below). It can be seen from Figure 4 that the conduction band of pure ZnO is mainly composed of the 4s state of Zn and the 2s state of O, and the valence band is mainly composed of the 3d state of Zn and the 2p state of O. The bottom of the conduction band and the top of the valence band are determined by the 4s state of Zn and the 2p state of O, respectively.

The energy band after doping is shown in Figure 5. The band gap of pure zinc oxide is 3.373 eV. After the Cu doping, the band gap becomes smaller. As the Cu doping amount increases, the band gap continues to decrease. This provides great possibilities for achieving high absorption of solar cell photoanodes in the visible range.

Combined with the partial density of states (Figure 6 and Figure 7), for pure zinc oxide in the conduction band part, the role of Zn-4s is not very large, and mainly depends on the Cu-3d and p electrons (O-2p, Cu-3p). With the incorporation of Cu^2+^, Cu 3d and p electrons also have a repulsive effect, causing the conduction band bottom energy to rise, so we see energy drop at the bottom of the valence band. The increase of the energy of the bottom of the conduction band will inevitably lead to an increase in the band gap. At the same time, as the concentration of Cu is increased, the electron concentration of Zn-3d in the valence band decreases, which further weakens the p-d rejection effect at the top of the valence band. With the increase of copper concentration, the concentration of Cu-3d and Cu-3p increased, and the concentration of O-2p did not increase, which also led to the enhancement of the p-d repulsion effect at the top of the valence band. This phenomenon is intensified by increased Cu doping, and the band gap width also changed. Due to the intervention of Cu^2+^ ions, the top of the valence band is moved upwards, the bottom of the conduction band is moved downwards, and the band gap becomes smaller. At the same time, as seen from Figure 5, the band structure appears near the Fermi level. The orbital hybridization phenomenon and a new energy level appear. At the symmetry point G, the −6.5 eV to −4 eV band hybridization phenomenon is more serious, while at −2.5 to −1.6 eV and −1.8 to −0.7 eV a new energy level also appears, and the localization is more obvious.

The number of valence electrons of copper is lower by one compared with zinc atoms. Therefore, copper doping is a p-type doping, which introduces holes in the valence band. It can also be seen from Figure 6 and Figure 7 that the Fermi level enters the valence band. The conduction band of pure zinc oxide is mainly composed of the 4s state of zinc and the 2s state of oxygen. The bottom of the conduction band is determined by the 4s state of zinc. The valence band is mainly composed of the 3d state of zinc and the 2p hybrid state of oxygen. At the top of the valence band, the 2p orbital interaction of oxygen produces anti-bonds between the s-like anti-bond and the p-like group, thereby forming a band gap. The calculation results show that with the increase of copper doping amount, the band gap of the copper single-doped system is narrowed, and the calculation results tend to be consistent with the experimental results [28].

It can be seen from Figure 6 and Figure 7 that the intervention of Cu causes localized density peaks near the Fermi level, and also promotes localization in the surrounding O-2p orbitals, while the surrounding Zn atoms are not subjected to Cu states. The effect of density peaks. This phenomenon indicates that Cu doping only affects the surrounding O atoms, and the localization characteristics are very prominent, which limits the electrons around Cu in moving only near the Cu–O bond. From Figure 5, it can be concluded for 3 × 3 × 2 (2.8%) that the band gap becomes smaller, which is 1.122 eV. Compared with Figure 6, it can be seen that the valence band becomes wider after doping, and the conduction band moves down at the same time. An impurity level is added between the strip and the conduction band, thereby making the band gap smaller, which creates conditions for realizing as much absorption of visible light as possible for the solar cell photoanode.

In this paper, the calculation method of GGA + U is used to take U = 5.0 eV for the Zn-3d state and U = 8.0 eV for the O-2p state in the doping system Cu_0.0278_Zn_0.9722_O, and the band gap was calculated by changing the the U value of Cu 3d. The result is shown in Table 2. It can be seen from Table 2 that on changing the 3d state U value of copper, the resulting band gap value is also very different, where in the band gap of the doping system it is at most 2.510 eV when U = 5 of the 3d state of copper, but it may be smaller than the experimental value.

### 3.3. Absorption Spectra Analysis

The absorption spectra of the pure ZnO unit cell, Zn_0.9722_Ag_0.0278_O, and Zn_0.9583_Ag_0.0417_O supercell in the wavelength range of 380–800 nm are shown in Figure 8 and Figure 9.

The pure zinc oxide absorption coefficient is less than 410 nm, indicating that zinc oxide is transparent in this range, and the absorption coefficient is small in the visible light region (400–790 nm). The photon energy of the zinc oxide light absorption edge is 410 nm, and the corresponding energy is 3 eV, which is the minimum energy that the electrons need to absorb from the top of the valence band to the bottom of the conduction band. As the wavelength increases, the absorption coefficient increases further, and there are multiple obvious absorption peaks in the 0–500 nm range. When the wavelength is 72 nm, the absorption coefficient reaches the maximum value. Combined with the DOS analysis, this is mainly the transition of O-2p and Zn-4s orbitals in the orbital guide bands of Zn-3d and O-2p with a relatively large density of states in the valence band. The absorption coefficient gradually decreases after the wavelength is greater than 72 nm.

The comparison of the light absorption coefficient of the pure zinc oxide and copper doped system (Figure 9) shows that for the pure zinc oxide in the visible region, the absorption coefficient is very low, and with increasing copper doping concentration, the light absorption edge of the copper substitution zinc is shifted towards the low energy direction, and the red shift phenomenon occurs. This is mainly because of the hole carriers generated after the copper doping, so that the Fermi level enters the valence band, which is consistent with the density of the states. It is also consistent with the experimental results [33,34]. At the same time, one can see that the absorption rate of the system in the ultraviolet–visible region is significantly increased after the incorporation of copper. As the doping concentration increases, the absorption rate becomes larger, which plays a certain role in developing transparent solar cell materials.

It can be seen from Figure 9 that in the doping amount range (*x* = 0, *x* = 0.0278, *x* = 0.0417), more Cu doping results in a more significant red shift of the absorption spectrum. The calculation results are consistent with the above-mentioned band gap width analysis, which is consistent with the experimental results [12].

At the same time, we prepared wurtzite-type zinc oxide and copper-doped zinc oxide by a two-step method. It can be seen from Figure 10 that the spectrum after copper doping has a large absorption in the visible light region, and the light absorption coefficient of visible light is obviously improved. It can be seen that the calculation and experimental results tend to be consistent. This indicates that copper doping increases the light absorption of the zinc oxide photoanode in the visible range, thereby increasing the photoluminescence yield and photocatalytic efficiency of zinc oxide, extending the spectral response range of zinc oxide, and improving the utilization of solar energy.

## 4. Conclusions

The effect of copper doping on the electronic structure and absorption spectrum of ZnO was studied by first-principles calculations and experiments. The conclusions can be drawn as follows:The band gap of pure zinc oxide calculated by GGA + U is 3.373 eV, which is consistent with the experimental values;The higher the Cu doping concentration, the larger the total energy, indicating that higher doping amount worsens the system stability;The incorporation of copper makes the impurity band and valence band of the ZnO system degenerate, and the conduction band bottom shifts to the low energy direction. After doping, the forbidden band width becomes smaller, so that the energy required for the electrons to make the transition from the valence band to the conduction band becomes smaller, and the light absorption undergoes a red shift phenomenon;The calculation results are consistent with the relevant experimental results. These comprehensive results show that copper-doped zinc oxide has better optical properties and the internal stress of the film is small, which can be used as a high-performance solar cell photoanode material.

## Figures and Tables

**Figure 1 materials-12-00196-f001:**
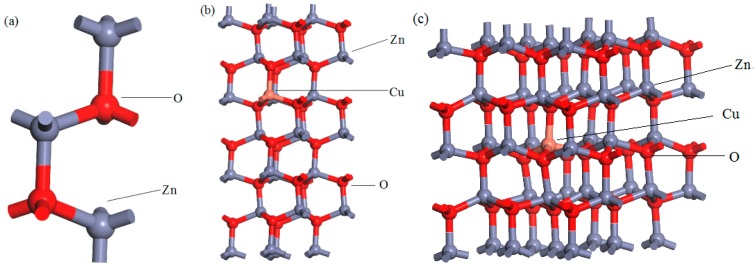
The models of: (**a**) pure ZnO, (**b**) Cu_0.0417_Zn_0.9583_O supercell, (**c**) Cu_0.0278_Zn_0.9722_O supercell.

**Figure 2 materials-12-00196-f002:**
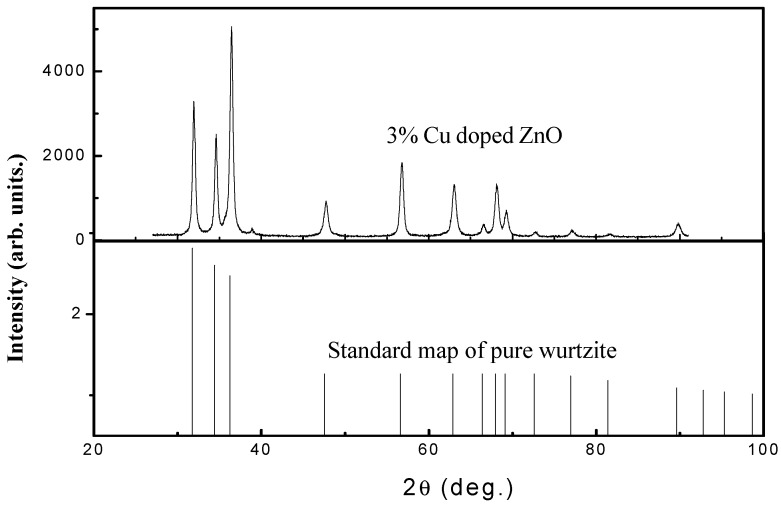
XRD patterns of Cu/ZnO (3% Cu) and standard pure wurtzite.

**Figure 3 materials-12-00196-f003:**
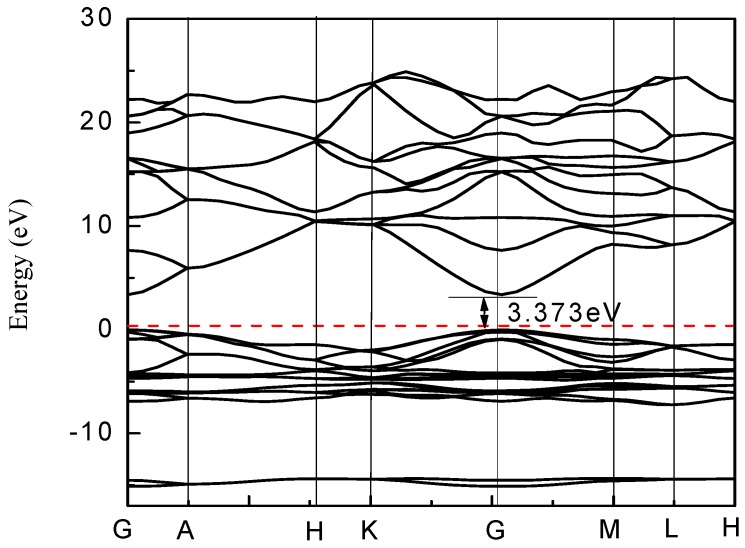
Band structure of pure ZnO.

**Figure 4 materials-12-00196-f004:**
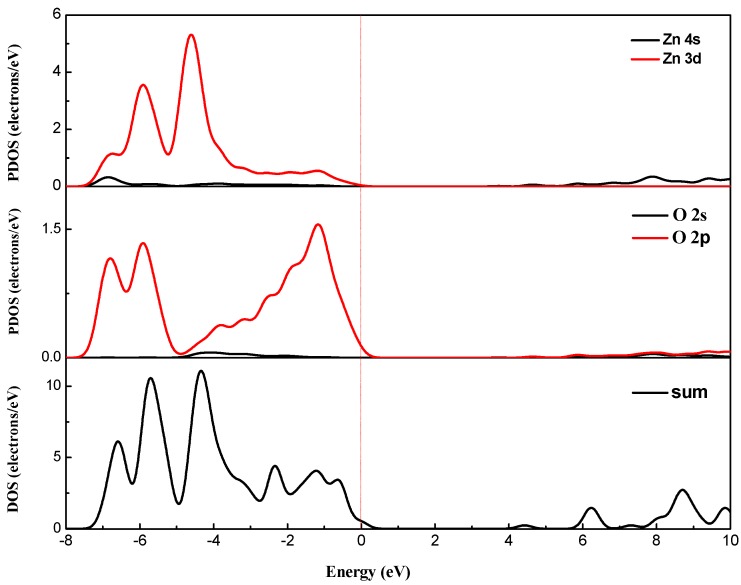
**Partial** density of states (PDOS) for pure ZnO.

**Figure 5 materials-12-00196-f005:**
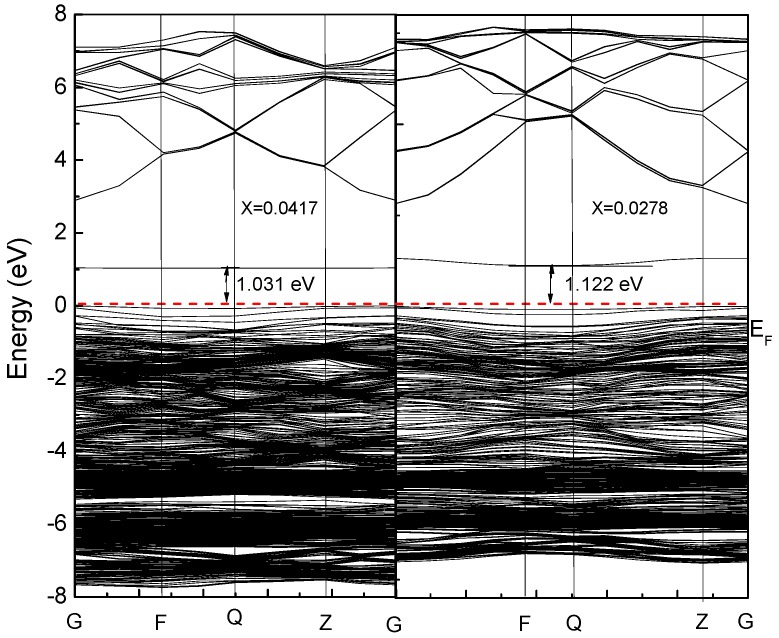
Band structure of Cu_x_Zn_1-x_O (x = 0.0417, 0.0278).

**Figure 6 materials-12-00196-f006:**
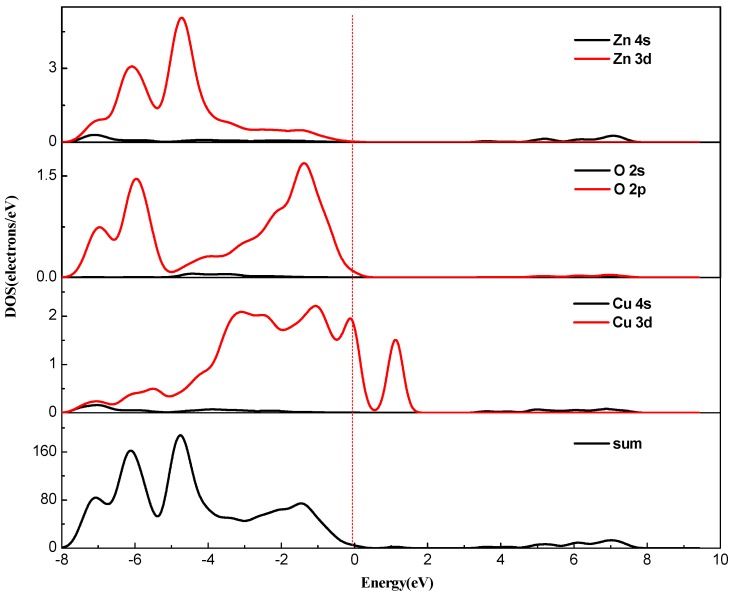
PDOS of Cu_0.0278_Zn_0.9722_O.

**Figure 7 materials-12-00196-f007:**
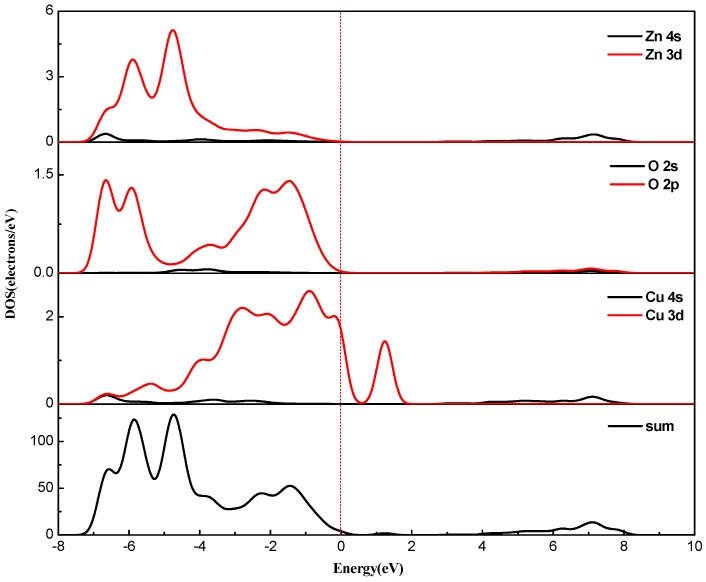
PDOS of Cu_0.0417_Zn_0.9583_O.

**Figure 8 materials-12-00196-f008:**
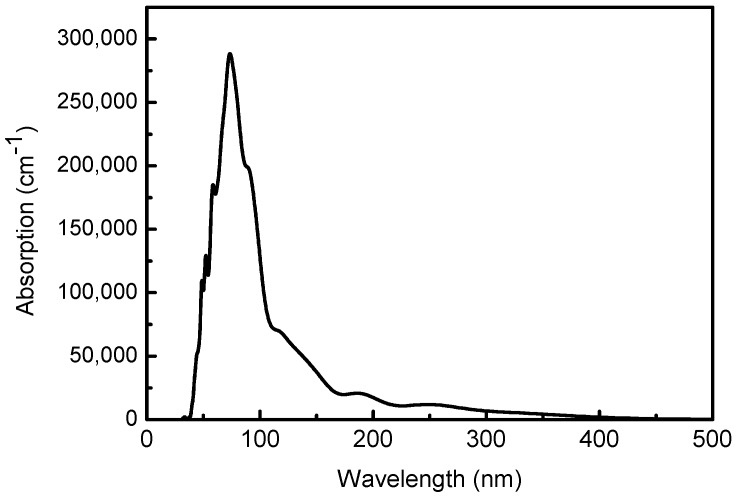
Absorption of pure ZnO.

**Figure 9 materials-12-00196-f009:**
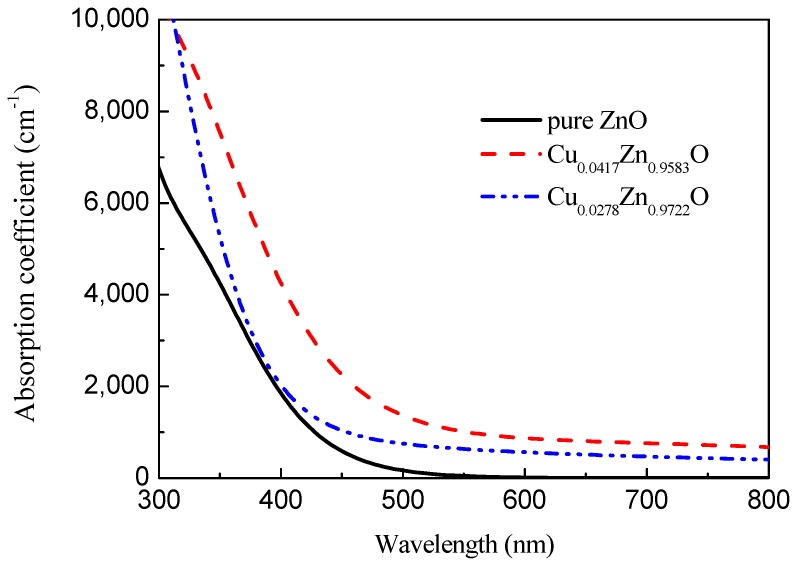
Absorption of Cu_x_Zn_1-x_O.

**Figure 10 materials-12-00196-f010:**
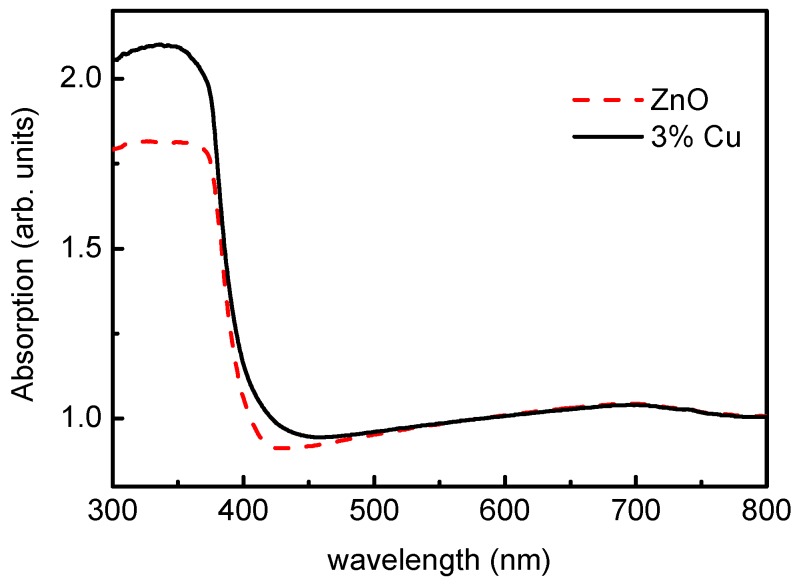
Absorption of pure ZnO and Cu doped ZnO.

**Table 1 materials-12-00196-t001:** The lattice parameters and formation energy of Zn_1-*x*_Cu*_x_*O_2_ after geometrical optimization.

Model	*a*, nm	*c*, nm	*V*, nm^3^	*c*/*a*	*E*_f_, eV
Cu_0.0417_Zn_0.9583_O	0.3272	0.5284	0.0612	1.5967	−2.24
Cu_0.0278_Zn_0.9722_O	0.3249	0.5205	0.0549	1.6020	−3.10
Pure ZnO	0.3249	0.5205	0.0549	1.6020	—

**Table 2 materials-12-00196-t002:** Band gap calculated by different U values.

		U_d,Cu_ (eV)	E_g_ (eV)	E_g_ (eV)
	ZnO	-	3.373	3.25 [29]
	Cu_0.0__278_Zn_0.97__22_O	2.5	1.031	-
		4	1.665	-
		5	2.510	-
		6	2.410	-
		7	2.203	-
Experiment value	ZnO	-	-	3.37 [30]
	Cu_0.02_Zn_0.98_O	-	-	3.18 [31]
	Cu_0.03_Zn_0.97_O	-	-	3.159 [32]

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
