# Peer review of "Cu-Doped ZnO Electronic Structure and Optical Properties Studied by First-Principles Calculations and Experiments"

_materials, 2019, doi:10.3390/ma12010196_

Round 1
Reviewer 1 Report
This work investigates the effect of Cu doping on the electronic properties of ZnO. The authors have calculated the impact of several doping levels and presented a thorough discussion on the electronics of the material. However, the paper has several issues and cannot be accepted in its present form. Below are some critical comments that must be addressed before the paper can be accepted.
1. The authors need to clarify what they meant by the word "trying" in the sentence "After trying, it is found that in all the systems the 3d state of Zn and the 2p state of O take U = 5 eV and U = 8 eV, respectively"? Was it a systematic calculation to determine the U values, or did they obtain the values from the literature?
2. What is the "principle of minimum energy in quantum mechanics"? I think the author might be pointing at the concept of thermodynamic equilibrium?
3. In the paragraph from line 117 to 123, the authors are just repeating themselves. It can be rephrased in only one sentence.
4. Line 124: "In this paper, Cu-doped ZnO was prepared by.." Which paper?
5. The XRD in Fig. 2: The authors only placed there without stating where it came from, whether it was calculated, and if so, how calculated? This information must be added to the paper.
6. The band structure figure is missing a figure number, and the two figures there are not aligned.
7. The authors are using a different x-axis units in Figure 7 and 8. The units should be the same.
Author Response
Point 1: The authors need to clarify what they meant by the word "trying" in the sentence "After trying, it is found that in all the systems the 3d state of Zn and the 2p state of O take U = 5 eV and U = 8 eV, respectively"? Was it a systematic calculation to determine the U values, or did they obtain the values from the literature?
Response 1: We removed the word “trying” in the revised paper. The sentence now reads: It is found that in all the systems the 3d state of Zn and the 2p state of O take U = 5 eV and U = 8 eV, respectively. Some calculations do not include this repulsion energy (U), the calculated band gap is small and its value is about 0.7 eV. This is because the traditional method of calculating the electronic structure underestimates the band gap width of the metal oxide, and some literature show that the 3d state of Zn and the 2p state of O take U = 5.5 eV and U = 8 eV, respectively, the band gap of the obtained pure zinc oxide is close to the theoretical value. We have also made many attempts. It is found that the 3d of state of Zn take U=5.5 is calculated to be small, when the 3d state of Zn and the 2p state of O take U = 5.5 eV and U = 8 eV, respectively, the band gap of zinc oxide is 3.373 eV, which is very close to the theoretical value (Section 3.2 have instructions on this).
Point 2: What is the "principle of minimum energy in quantum mechanics"? I think the author might be pointing at the concept of thermodynamic equilibrium?
Response 2: The principle of the least action. In thermodynamics, the more stable the system, the lower its energy state. Corresponding modifications were made in the revised paper.
Point 3: In the paragraph from line 117 to 123, the authors are just repeating themselves. It can be rephrased in only one sentence.
Response 3: We have made corresponding modifications
Point 4: Line 124: "In this paper, Cu-doped ZnO was prepared by.." Which paper?
Response 4: Cu-doped ZnO was prepared by our group. The paper has been added for the preparation method and detection means.
Point 5: The XRD in Fig. 2: The authors only placed there without stating where it came from, whether it was calculated, and if so, how calculated? This information must be added to the paper. Response 5: This information is now added to the revised paper.
Point 6: The band structure figure is missing a figure number, and the two figures there are not aligned.
Response 6: We have modified the figure.
Point 7: The authors are using a different x-axis units in Figure 7 and 8. The units should be the same.
Response 7: We have modified Figure 7, and now the coordinates of the two curves are the same.

Reviewer 2 Report
1. In this manuscript, structural, electronic and optical properties of Cu doped ZnO were studied both theoretically and experimentally.
Theoretical calculations were based on density functional theory. Supercells with two different doping concentrations (2.8 and 4.2 %) of Cu were constructed and then optimized. Electronic band structures and (partial) density of states were used to study the electronic properties. Then calculated absorption coefficients were shown. Experiments were performed for 3% Cu doped ZnO. X-ray diffraction pattern and absorption spectra were presented.
2. I do not recommend to publish this manuscript, because it lacks in novelty. There were already a lot of publications, which studied structural, electronic and optical properties of Cu-doped ZnO both experimentally and theoretically, some of which are cited in this manuscript (e.g. Refs. [5] and [10]), and no new results were displayed in the manuscript.
3. It is already well-known that Cu-doped ZnO shows band gap narrowing, which results in the red shift of the absorption edge. In addition, the manuscript has some problems, e.g. For the experiment, there are no details about how XRD pattern and absorption spectra were measured. (In addition, Figure 9 looks like transmission spectra rather than absorption spectra.)
4. For the partial density of states, I do not understand how Cu 3p states and Zn 3p states were obtained, although the pseudopotentials mentioned in the section 2.1 did not include these electrons as valence.
5. For the band structure (Figure number is missing), the scale along the y-direction is different between right and left panel, which exaggerates the difference of the band structures between different doping concentrations.
Author Response
Response to Reviewer 2 Comments
Point 1: In this manuscript, structural, electronic and optical properties of Cu doped ZnO were studied both theoretically and experimentally. Theoretical calculations were based on density functional theory. Supercells with two different doping concentrations (2.8 and 4.2 %) of Cu were constructed and then optimized. Electronic band structures and (partial) density of states were used to study the electronic properties. Then calculated absorption coefficients were shown. Experiments were performed for 3% Cu doped ZnO. X-ray diffraction pattern and absorption spectra were presented. I do not recommend to publish this manuscript, because it lacks in novelty. There were already a lot of publications, which studied structural, electronic and optical properties of Cu-doped ZnO both experimentally and theoretically, some of which are cited in this manuscript (e.g. Refs. [5] and [10]), and no new results were displayed in the manuscript.
Response 1: Other papers demonstrate a lot of Cu-doped ZnO calculations, but most of them focus on the magnetic and optical properties. When compared with the experimental results, most of them are concentrated on other methods to prepare doped zinc oxide, not the two steps method. The two-step method has low cost of raw materials, equipment requirements are simple and good performance has been demonstrated to prepare ZnO nano-crystalline powder. Compared with other methods, such as magnetron sputtering, hydrothermal method has many advantageous characteristics, such as dosage of doping elements.
Point 2: It is already well-known that Cu-doped ZnO shows band gap narrowing, which results in the red shift of the absorption edge.
Response 2: In most reported experiments the absorption band had a red shift, but also a small amount of blue shift was also reported. In this paper the preparation method is relatively uncommon, namely the two-step method, which can accurately control the amount of dopant, demonstrating more consistent results.
Point 3; In addition, the manuscript has some problems, e.g. For the experiment, there are no details about how XRD pattern and absorption spectra were measured. (In addition, Figure 9 looks like transmission spectra rather than absorption spectra.)
Response 3: More details about the experimental method were added. The energy band diagram was also added.
Point 4: For the partial density of states, I do not understand how Cu 3p states and Zn 3p states were obtained, although the pseudopotentials mentioned in the section 2.1 did not include these electrons as valence.
Response 4: The calculation method of the Cu 3p states and the Zn 3p states is the same as Cu 3d4s states and Zn 3d4s states. The pseudopotentials did not include these electrons as valence electrons. We have modified the figure and deleted the Zn 3p states in the figure. Band structure is affected by the repulsion between 3p and 3d of Cu, so there is 3p in the Cu figure.
Point 5: For the band structure (Figure number is missing), the scale along the y-direction is different between right and left panel, which exaggerates the difference of the band structures between different doping concentrations.
Response 5: We have modified the figure. In pure zinc oxide band diagram the ordinate range is larger, which is why we don't narrow the scale of the ordinate. The purpose is to present a complete band diagram, after doped ZnO band diagram where there are many gaps in the wide range, compared with a lot of literature reports. Copper doping can illustrate this relatively small range, so we can narrow the ordinate.

Round 2
Reviewer 1 Report
The authors have reasonable addressed the referee's comments, except for the first comment. It is still not clear why the authors chose to those two values? And I recommend that the authors cite a reference on the U values from the literature. Finally: the authors said that the band gap without U gives 0.7 eV, I think this is wrong, the GGA band gap is ~2 eV.
Author Response
Dear reviewer: Thank you for your review of my manuscript, I have benefited a lot, thank you for your guidance!
Point 1: The authors have reasonable addressed the referee's comments, except for the first comment. It is still not clear why the authors chose to those two values? And I recommend that the authors cite a reference on the U values from the literature. Finally: the authors said that the band gap without U gives 0.7 eV, I think this is wrong, the GGA band gap is ~2 eV.
Response 1: Thank you for your advice. The U value select in the paper does from the literature. On this basis, we have made calculations and obtained a more suitable u value. I have already introduced references in the paper. In the literature, the calculated band gap of pure zinc oxide is 0.792 eV. (Jun-Qing Wen, Jian-Min Zhang, Ze-Gang Qiu, The investigation of Ce doped ZnO crystal: The electronic, optical and magnetic properties, Physica B: Condensed Matter 534 (2018) 44– 50). The U values for undoped ZnO are Ud,Zn = 10:00 eV and Up;O = 7:00 eV in the LDA+U method. and the calculated bandgap of the undoped ZnO is 3.40 eV (Qing-Yu Hou, Effect of heavy Ag doping on the physical properties of ZnO, International Journal of Modern Physics B 2018, 32: 1850099 (19 pages)). The Coulombic energy U of Zn-3d, O-2p were set to 10 eV, 7 eV, the band gap for undoped ZnO is 3.40 eV after using the GGA+U method. ( Yong Li, Study on electrical structure and magneto-optical properties of W-doped ZnO, Journal of Magnetism and Magnetic Materials , 2018, 451:697–703). The Coulombic energy U of Zn-3d, O-2p were set to 5.5 eV, 8.0eV was reasonable. (Guo Shao-Qiang, First principles study of the effect of high V doping on the optical band gap and absorption spectrum of ZnO,Acta Phys. Sin. 2014 , 63, (10): 107101)).

Reviewer 2 Report
The manuscript has improved after the revision. However, it still has a number of problems for publication. I find novelty of this work only in experimental side. The manuscript should be rewritten to focus on experiment.
Also, In Response 1, it is written that "but most of them focus on the magnetic and optical properties". This is true (for example, Horzum et al., Philos. Mag. 96, 1743 (2016), Xu et al., J. Appl. Phys. 105, 043708 (2009)) and the present manuscript also focuses on optical properties. So I still do not find any novelty in calculated part of the manuscript.
In Response 2, it is written that "also a small amount of blue shift was also reported". The blue shift was observed in the higher doping case, such as 15% - 20% doping. In the lower doping case (5% or below) only red shift was observed. There is no inconsistency in literature.
In Response 3, the problem of Figure 10 (former Figuere 9) was not addressed. If the figure corresponds to the absorption, ZnO would have high absorption in visible light range without doping and furthermore doping would lower the absorption. This contradicts what we know. Instead, the figure is similar to transmission spectra shown in Figure 4 in Philo. Mag. 96, 1743 (2016) paper and Fig. 4 in Appl. Surf. Sci. 270, 104 (2013) paper.
In Response 4, Cu 3d hybridizes with O 2p, not Cu 3p. See, for example, Ferhat et al., Appl. Phys. Lett. 94, 142502 (2009), Ghajari et al., J. Magn. Magn. Mater. 325, 42 (2013).
Author Response
Dear reviewer:
Thank you for your review of my thesis, and I have benefited a lot by modifying the paper. Thank you very much!
Point 1: The manuscript has improved after the revision. However, it still has a number of problems for publication. I find novelty of this work only in experimental side. The manuscript should be rewritten to focus on experiment. Also, In Response 1, it is written that "but most of them focus on the magnetic and optical properties". This is true (for example, Horzum et al., Philos. Mag. 96, 1743 (2016), Xu et al., J. Appl. Phys. 105, 043708 (2009)) and the present manuscript also focuses on optical properties. So I still do not find any novelty in calculated part of the manuscript.
Response 1: Thank you for your advice, However, there is a key test in the experimental data because the instrument failure has not been completed, so it can only correspond to the performance of the calculation part. For the calculation part, I think the highlight of the paper is that the band gap of pure zinc oxide calculated by the GGA+U method is close to theoretical value, the value of U has a certain reference for future calculations. Of course, the paper is indeed the highlight of our highlights. I have modified the paper, added the calculation of the U value of copper, and the experimental value. Comparison.
Point 2: In Response 2, it is written that "also a small amount of blue shift was also reported".The blue shift was observed in the higher doping case, such as 15% - 20% doping. In the lower doping case (5% or below) only red shift was observed. There is no inconsistency in literature.
Response 2: In most reported experiments the absorption band had a red shift, but also a small amount of blue shift was also reported. It is true that when the copper doping is below 5%, there is more redshift. In literatures(Luo Shuang,Correlations between optical band gap and magnetism performance of Cu-doped ZnO,Chemical Engineer,2015,(12):11), it was observed that when the copper doping was less than 5%, there is a blue shift.
In literatures(Li Jianchang, Growth and Characterization of Cu-Doped ZnO Sol-Gel Films, Chinese Journal of Vacuum Science and Technology, 2012,32(3):238) it was also observed that when the copper doping was less than 1%, there is a blue shift.
Point 3: In Response 3, the problem of Figure 10 (former Figuere 9) was not addressed. If the figure corresponds to the absorption, ZnO would have high absorption in visible light range without doping and furthermore doping would lower the absorption. This contradicts what we know. Instead, the figure is similar to transmission spectra shown in Figure 4 in Philo. Mag. 96, 1743 (2016) paper and Fig. 4 in Appl. Surf. Sci. 270, 104 (2013) paper.
Response 3: Thank you for your correction. Maybe there was a problem with the parameter setting of the instrument. The data is indeed abnormal. But the result is not the transmission spectrum. We have re-executed the experiment. Figure 10 has been replaced with the result of the new work.
Point 4: In Response 4, Cu 3d hybridizes with O 2p, not Cu 3p. See, for example, Ferhat et al., Appl. Phys. Lett. 94, 142502 (2009), Ghajari et al., J. Magn. Magn. Mater. 325, 42 (2013).
Response 4: I am very sorry to understand the content of the information, I have modified the density state figure, thank you for your correction.

Round 3
Reviewer 2 Report
The authors made corrections and further improved to the manuscript. Now it can be published after following minor change.
In the added paragraph from line 215 to line 224. The latter half, "It may be that ..." to the end of paragraph does not make sense to me. You need to do calculations with different charge state to model different valence states. In this case, you need to add one electron to the system to simulate Cu+ state. Such calculations may be beyond the scope of this paper, so I recommend to simply remove this part (from l. 220, "It may be that ..." to l. 224 "... affect the band gap").
Author Response
Dear reviewer:
Thank you again for your review of my manuscript.
Point 1: Comments and Suggestions for Authors: The authors made corrections and further improved to the manuscript. Now it can be published after following minor change.
In the added paragraph from line 215 to line 224. The latter half, "It may be that ..." to the end of paragraph does not make sense to me. You need to do calculations with different charge state to model different valence states. In this case, you need to add one electron to the system to simulate Cu+ state. Such calculations may be beyond the scope of this paper, so I recommend to simply remove this part (from l. 220, "It may be that ..." to l. 224 "... affect the band gap").
Response 1:Thank you for your suggestion, I have removed this part.
